# Verification of Fuel Consumption and Carbon Dioxide Emissions under Sulfur Restriction Policy during Oceanographic Navigation

Hsueh-Chen Shen [1], Fu-Ming Tzu [2], Chitsan Lin [1,3,*], Chin-Ko Yeh [2], Wen-Yen Huang [3], Han-Pin Pu [2] and Shun-Hsyung Chang [4]

1 Ph.D. Program in Maritime Science and Technology, National Kaohsiung University of Science and Technology, Kaohsiung 81157, Taiwan
2 Department of Marine Engineering, National Kaohsiung University of Science and Technology, Kaohsiung 81157, Taiwan
3 Department of Marine Environmental Engineering, National Kaohsiung University of Science and Technology, Kaohsiung 81157, Taiwan
4 Department of Microelectronics Engineering, National Kaohsiung University of Science and Technology, Kaohsiung 81157, Taiwan
* Correspondence: ctlin@nkust.edu.tw; Tel.: +866-7-3617141; Fax: +886-7-3651472

**Abstract:** The paper presents a comparison of the fuel oil (FO) consumption and carbon dioxide ($CO_2$) emissions of a container ship's 8000 twenty-foot equivalent unit (TEU) during oceanographic navigation. The evaluation has two types of FOs: a 3.4% heavy fuel oil with desulfurization (HFOWD) and a 0.5% very-low-sulfur fuel oil (VLSFO), based on the sulfur cap policy of the International Maritime Organization (IMO). The results show the average FO consumption at 130 tons/day of HFOWD and 141 tons/day of VLSFO, which means shifting to VLSFO increases fuel consumption 8.4% more than the HFOWD. The average $CO_2$ emissions are 429 tons/day of the HFOWD and 471 tons/day of the VLSFO, indicating an 9.5% increase in $CO_2$ emissions when the IMO adopts the low-sulfur fuel policy. Moreover, the VLSFO blending of various chemicals further deteriorates and wears out the main engine of the ship. IMO's low-sulfur fuel policy significantly reduced the emission of sulfur oxides ($SO_X$) and particulate matter emissions. Still, we should not ignore the fact that adopting VLSFO may cause more $CO_2$ emissions. Therefore, while switching to low-sulfur fuels, the maritime industry should improve the related energy efficiency to reduce fuel consumption and $CO_2$ emissions.

**Keywords:** heavy fuel oil with desulfurization (HFOWD); very-low-sulfur fuel oil (VLSFO); fuel oil consumption; carbon dioxide ($CO_2$) emissions; greener marine industry; global warming

## 1. Introduction

Seaborne cargo is the most popular type of transportation globally, accounting for 90% of global trade [1,2]. Consequently, the emission of pollutants from active ships is undoubtedly deteriorating the marine environment. The IMO held a conference on greenhouse gas (GHG) in 2020 [3,4], indicating that GHG emissions consist of $CO_2$, methane ($CH_4$), and nitrous oxide ($N_2O$) from global shipping. Moreover, the sixth assessment report released by the Intergovernmental Panel on Climate Change (IPCC) pointed out that the average annual greenhouse gas emissions from 2010 to 2019 were higher than ever [5]. Global greenhouse gas emissions are expected to peak before 2025 [6]. Among the GHGs, $CO_2$ emissions increased significantly and are deemed a notorious gas. To reduce the pollution and impact of the global shipping industry on the environment, the IMO committed to reduce total GHG emissions from shipping by at least 50% before 2050 [7–9].

The air pollutants of ships arising from fuel consumption are due to the main engine, generator, and boiler. Both the main engine and the generator are diesel engines. Sulfur

content in FO affects the composition and total amount of exhaust gas. The hydrocarbon content in the FO is essential to determine the $CO_2$ emission. It is known that the heating value of FO also affects $CO_2$ emissions. The $CO_2$ emissions of a ship are directly proportional to its fuel consumption, which is determined by the engine efficiency. At the same time, the $CO_2$ emissions are proportional to the output power of the engine, also known as the engine load. On the other hand, sulfur oxide ($SO_X$) emissions cause acid rain to endanger the land, ocean, and human health.

Table 1 indicates the chronological evolution of maritime FO, which has varied over the past two decades.

**Table 1.** Chronological evolution of IMO fuel sulfur regulations in maritime industry.

| Effective Date | Global Area Fuel Sulfur Limits | Emission Control Areas (ECAs) Fuel Sulfur Limit |
| --- | --- | --- |
| 19 May 2005 | <4.5% m/m | <1.5% m/m, Baltic Sea region and North Sea Region |
| 1 January 2010 | | <1.0% m/m, Baltic Sea region and North Sea Region |
| 1 January 2012 | <3.5% m/m | 1 August 2012, With North America included |
| 1 January 2014 | | With US Caribbean included |
| 1 January 2015 | | <0.1% m/m, including the Baltic Sea, the North Sea, North America, and the Caribbean |
| 1 January 2020 | <0.5% m/m | |
| 1 March 2020 | <3.5% m/m + desulfurization | |

In the literature review, Hakoun et al. (2021) [10] analyzed the $CO_2$ emission coefficient of various fuels with different sulfur content by using the data of shipping records from Asia to North America from 2010 to 2017. The result indicate that the scrubber is the most mature technical solution to reduce the sulfur pollution emissions in liner shipping using the 3.5% HFO. Peng (2016) [11] utilized the tribological property of pure petrol–diesel and studied the ring wear tester. As a result, the evaluation shows that a low concentration of biodiesel blends is more effective as a lubricant because of the polarity. Narayan et al. (2018) [12] investigated the idea that small amounts of ultralow fuel oil sulfur may damage engine parts due to poor lubrication and decreased coalescence ability. Norouzi et al. (2014) [13] investigated the effect of the corrosive behavior in blends with ultra-low-sulfur diesel in the bi-metal part of the diesel engine for aluminum and copper. The test temperature was 80 degrees Celsius during the operation time of 600 to 5760 h. As a result, a higher tendency for corrosion is shown in the degradation of copper compared with aluminum. The degradation of copper metal is due to the lower resistance of the formed oxide layer on the surface. Consequently, the corrosion behavior is more severe than in aluminum. Such findings raise negative concerns about adopting ultra-low-sulfur diesel FO. Nevertheless, Yeh et al. (2022) [14] recognized that the IMO's low-sulfur fuel policy has significantly reduced the emission of sulfur oxides and the quantity of particulate matter.

The existing problem is the high emission of $CO_2$ in the maritime industry. Thus, choosing the appropriate FO is a priority task, since global shipping mainly adopts three types of marine FOs. Before 2019, the number of users in descending order was HFO > low-sulfur fuel oil (LSFO) > liquefied natural gas (LNG). Among the FOs, it was found that HFO consumption was the most effective and preferred for shipping [15–17], followed by LSFO and LNG. After January 1 2020, the number of users shifted, changing the order to VLSFO > HFOWD > LNG [18,19]. However, VLSFO suffers from higher costs, and the LNG is the most expensive among the above FOs. Still, we should not ignore that adopting VLSFO may cause more $CO_2$ emissions. From March 1 2020, the IMO allowed the shipping industry to keep using HFO. However, if a ship chooses the HFO, adding a desulfurization device is required; this is HFOWD. On the other hand, the characteristic of VLSFO is that it can be used as fuel without adding a desulfurization device or replacing the hosting engine. As VLSFO is a newly developed fuel from the traditional supply chains, whether a stable, qualified supply can be guaranteed raises another concern [14].

To address the above-mentioned $CO_2$ emission concerns, this paper compares two candidate fuel oils to evaluate a ship's practical fuel consumption and $CO_2$ emission during oceanographic navigation; these fuels oils are HFOWD and VLSFO. The measurement on the board is under the sulfur restriction policy during a voyage of 18 months in the Pacific Ocean. Consequently, the analysis is significant to the marine industry in mitigating carbon dioxide emissions and global warming effects.

## 2. Materials and Methods

The experiment utilizes two identical container ships to verify the FO consumption and the $CO_2$ emissions under the same environmental conditions, mainly in the transoceanic route between Asia and Central America (C.A.). The first container ship utilized HFOWD on the route (as shown in Figure 1, the red dotted line). The other container ship without a desulfurization device, adopted VLSFO on the route (as shown in Figure 1, the blue dotted line). Both ships sailed the route in the ocean across18 months, from July 2020 to February 2022. The experiment acquired the measurement in the open ocean (Figure 1, the solid green line segment). The data collection of the experiment was during similar seasons, sea water temperature, and weather conditions during the voyages.

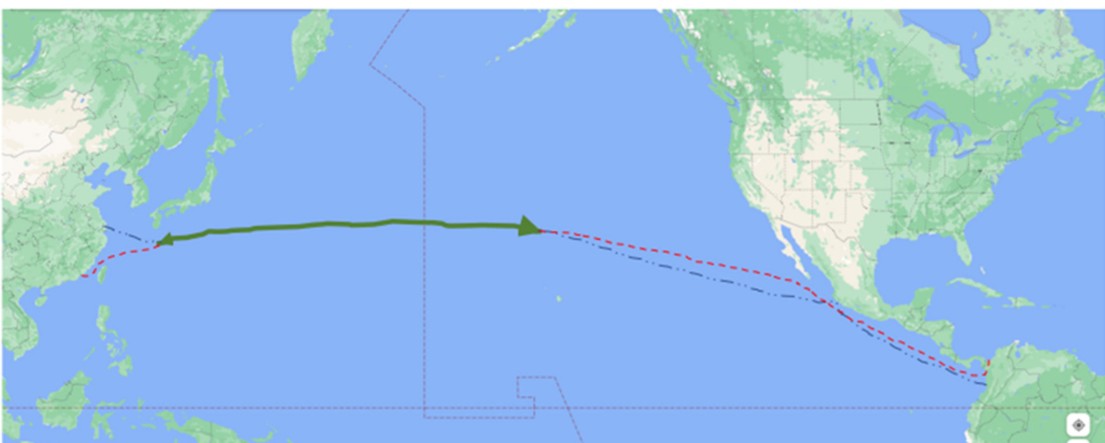

**Figure 1.** Transoceanic routes measured the FO consumption and carbon dioxide emissions in the solid green line segment.

The red and green dotted line voyages were routed back and forth from Asia to C.A. through the Pacific Ocean. The solid green line segment is the route close to the Asian international waters, where the data collection took place in the ocean. The fuel consumption depends on the output power of the main engine. The sea water temperature, weather conditions, tide situation, ship speed, etc., affected the output power of the main engine.

### 2.1. Container Ship Description

The experiment was carried out on two oceangoing container ships of the same engine type. Among global container ships, the loading capacity of 7500 to 9999 TEU is the most significant in maritime industry, and the ships we examined adopted the above 8000 TEU. The detailed specification of the container ship is in Table 2. The HFOWD container ship installed an advanced marine emissions control system (AMECS) and an exhaust gas cleaning system (EGCS) made by Wartsila Corporation (FI-00080, WÄRTSILÄ, Helsinki, Finland), which is an open-loop scrubber EGC system. The certification of this scrubber complied with IMO resolution MEPC 259 guidelines for exhaust gas purification systems. The ship also installed a continuous emission monitoring system (CEMS) to ensure the emission value of scrubbing water complied with the regulations and the sulfur oxides emitted to sulfur-containing fuel. The other container ship without desulfurization device adopted the VLSFO. The operation of the main engine in the experiment adopts a digital mode, and its output power is by digital signal, which transmits to the engine control room

(ECR) during the accuracy measurement while the engine is in operation. In addition, the original provider regularly performed the on board calibration to check on the equipment's accuracy. Table S1 shows a previous result that tabulates the consistency with the engine output, rpm, and fuel oil consumption in this investigation.

**Table 2.** Technical specifications of the container ships.

| Item | Description | Unit |
| --- | --- | --- |
| Launched date | 2013 | - |
| Length | 334.8 | Meter |
| Breadth | 45.8 | Meter |
| Draft design | 13.5 | Meter |
| Capacity | 8000 above | TEU |
| Main engine | MAN B&W 9K98ME Mark 7.1<br>MCR: 56~80% (77~88 RPM)<br>NCR: 50,463 kW and shaft speed 93.7 RPM | - |
| Gross tonnage | 99,998 | Tonnage |
| Service speed | 24.5 | knot |

Note: MCR—maximum continuous rating and NCR—normal continuous rating.

### 2.2. Estimation for FO Consumption and $CO_2$ Emissions

The experiment also investigated the engine to record the daily seawater temperature, power output, and fuel consumption. The $CO_2$ emission was calculated according to fuel consumption. The conversion of $CO_2$ emissions refers to the guidelines for national greenhouse gas inventory issued by the IPCC [20,21]. As shown below, $CO_2$ emissions from fuel combustion are in Equation (1).

$$CO_2 \text{ Emissions} = \sum_{\text{all fuels}} \left[ ((AC_{Fuel} \times CF_{Fuel} \times CC_{Fuel}) \times 10^{-3} - EC_{Fuel}) \times COF_{Fuel} \times 44/12 \right] \quad (1)$$

where $AC_{Fuel}$ indicates the daily fuel consumption as tons of the unit. As the below equation shows:

$$AC_{Fuel} = \text{Volume} \times Sg \times 1.006 \quad (2)$$

Density correction, which $Sg$ indicates, corresponds to the recorded operational temperature on board. As the below equation shows.

$$Sg = \text{Density} \times \{1 - [(T\,°C - 15\,°C) \times 0.00065]\} \quad (3)$$

The $V$ is the fuel oil volume (m$^3$) recorded on board. It expresses the specific gravity (mt/m$^3$) corresponding to the recorded operational temperature on board, and the coefficient of 1.006 is the oil purifier [22]. The density indicates the mass per unit fuel volume at 15 °C (mt/m$^3$). T °C is the meaning of the temperature of FO after heating, while 0.00065 is a coefficient [23]. The conversion factor $CF_{Fuel}$ is converted according to the typical calorific value in international energy statistics of the International Energy Agency (IEA). At the same time, $CC_{Fuel}$ means the carbon content in fuel and heavy oil is 21.1 [24,25], and the unit is kg C/GJ. Both HFO and VLSFO are residual fuel oils. According to the National GHG inventory guidelines issued by IPCC (2006) [20,21], a residual fuel oil's carbon dioxide emission factor ($CO_2$ emission factors) is 77,400 kg/TJ. They are consistent with the same carbon content. The $EC_{Fuel}$ carbon in feedstocks and non-energy use should be excluded from fuel combustion emission; it is zero in the study scenario. The $COF_{Fuel}$ means the carbon oxidation factor [26,27], which refers to the proportion of carbon that is oxidized. This value is 1, indicating complete oxidation, and 44/12 is a ratio of the molecular weight ratio of $CO_2$ and C.

Furthermore, the $CO_2$ emission indicates a calculation of the fuel consumption, also known as mass at kilograms (kg). The volume is a flowmeter to gauge the FO at m$^3$. The flowmeter measuring equipment utilizes the VAT, which is a professional instrument from France. The accuracy is at 0.1% deviation. Figure S1 provides the flowchart illustration and

Figure S2 exhibits an example gauge. The purifier is used by centrifugal force to separate the oil, moisture, and impurities with different specific gravity to obtain a proper FO. Before the FO is delivered to the purifier, the oil quality appears poor, with impurities that cannot be injected into the internal engine. After passing through the purifier, the FO quality's appearance tends to be a helpful oil, due to the high-speed rotation of the bowl disc [28].

This experiment was conducted on "IBM SPSS statistics 20" software for statistical data analysis. The software was carried out with mean ± standard deviation. Independent sample *t*-test was used to compare scheme one with HFOWD with an open-loop desulfurization tower and scheme two with VLSFO for the difference between fuel consumption and $CO_2$ emissions. The 95% confidence interval (CI) was used as the decision-making index in the statistical verification. The *p*-value < 0.05 was used as the criterion of statistical significance. In statistics, the 68–95–99.7 rule is the percentage of a normal distribution within one, two, or three standard deviations from the mean value.

Furthermore, *p*-value is a probability that takes a value between 0 and 1, between possible and impossible. So, suppose the *p*-value is 5%. In that case, the confidence interval is 95% (two combined = 1), reflecting that it is highly correlated with reality. Therefore, the *p* value is used to measure the strength of experimental evidence to support a conclusion.

## 3. Results and Discussion

### 3.1. Description Analysis for Both Fuel Options

The experiment was conducted on a practical ship while transoceanic in the Pacific Ocean. These ships in the ocean sailed back and forth for 18 months, from Asia to C.A. The investigation measures the two fuel options, HFOWD and VLSFO, to evaluate the FO consumption and the $CO_2$ emissions. As shown in Table 3, the main engine output power was recorded on board. The specific gravity of fuel after heating was calibrated by Equation (3). The fuel oil suppliers provided lower heating values (LHV). The main engine fuel consumptions were calculated according to Equation (2), and the $CO_2$ emissions were calculated from Equation (1).

**Table 3.** Comparison of the parameters for HFOWD and VLSFO.

| Voyage Route | Measurement | M (HFOWD) | M (VLSFO) | *t* | *p* |
|---|---|---|---|---|---|
| Asia to C.A. | Main engine output power (%) | 59.37 | 58.54 | 0.368 | 0.715 |
| | Specific gravity of fuel after heating (mt/m$^3$) | 0.94 | 0.933 | 4.795 | 0.000 |
| | Lower heating value (LHV) (kcal/kg) | 10,210 | 10,294 | −4.807 | 0.000 |
| | Main engine fuel consumption (ton/day) | 131.63 | 142.32 | −1.837 | 0.074 |
| | Main engine $CO_2$ emission (ton/day) | 435.18 | 474.8 | −2.084 | 0.044 |
| C.A. to Asia | Main engine output power (%) | 53.7 | 54.3 | −0.287 | 0.776 |
| | Specific gravity of fuel after heating (mt/m$^3$) | 0.943 | 0.933 | 6.438 | 0.000 |
| | Lower heating value (LHV) (kcal/kg) | 10,180 | 10,287 | −6.418 | 0.000 |
| | Main engine fuel consumption (ton/day) | 128.05 | 140.09 | −2.223 | 0.031 |
| | Main engine $CO_2$ emission (ton/day) | 421.98 | 467.019 | −2.564 | 0.014 |

Though the two voyages (HFOWD and VLSFO) are the same type of container ship and on the same transpacific routing, one may still question the comparability. Therefore, it is essential to demonstrate the issue. From Equations (1) and (2), $CO_2$ emission was calculated from fuel consumption corresponding to the main engine output power. Thus, if the main engine output power of the two voyages is similar (no significant difference), the two voyages can be compared.

Table 3 compares the navigation data acquired when the ships sailed from Asia to C.A. The mean values of the main engine output power are 59.37% and 58.54% of the *p* valve at 0.715, respectively. The statistic shows no significant difference in the output power between the two FO options. Similarly, while sailing from C.A. back to Asia, the mean values of the main engine output power pairs show the *p*-valve at 0.776 (>0.05), indicating no significant difference. Therefore, we can assume that the two voyages are comparable in

discussing the $CO_2$ emission difference. The raw data the voyages provided are shown in Table S2.

### 3.2. Fuel Consumptions and $CO_2$ Emissions for Both Fuel Options

The 8000 TEU container ships' travelled on round trips back and forth between Asia and C.A. As Table 3 indicates, the specific gravity of the fuel is at 0.940 mt/m$^3$ and 0.933 mt/m$^3$ for HFOWD and VLSFO, respectively ($p = 0.000$). The lower heating value is 10,210 kcal/kg and 10,294 kcal/kg, respectively ($p = 0.000$). The specific gravity of fuel ($p = 0.000$), the heating value ($p = 0.000$), daily fuel consumption ($p = 0.031$), and the daily $CO_2$ emission ($p = 0.014$) of the main engine in the HFOWD and VLSFO from C.A. to Asia are statistically significant.

Figure 2a indicates that the main engine fuel consumption from Asia to C.A. is 131.63 tons/day and 142.32 tons/day for HFOWD and VLSFO, respectively. Thus, there is a difference of 10.69 tons/day (7.8%). It turns out that HFOWD saves more fuel oil than VLSFO. Similarly, the main engine fuel consumption from C.A. to Asia is 128.05 tons/day and 140.09 tons/day for HFOWD and VLSFO, respectively, meaning there is a difference of 12.04 tons/day (9%). The HFOWD can save 11.36 tons/day (8.4%) of fuel compared with the VLSFO.

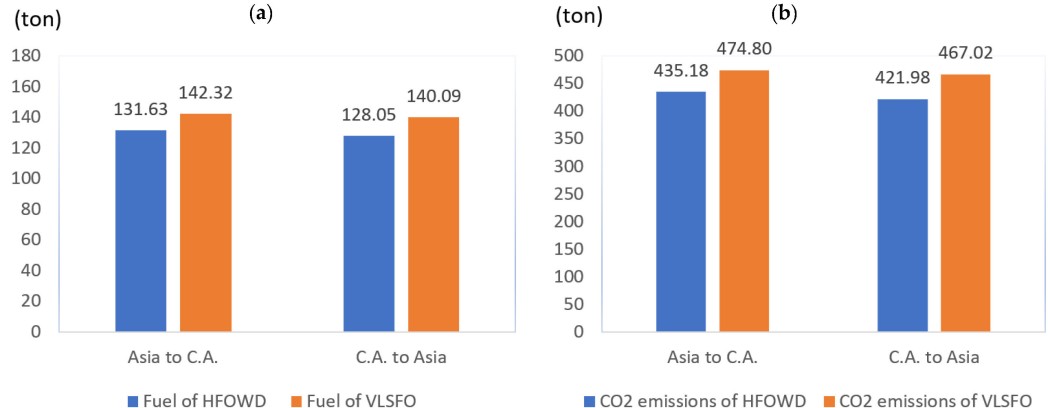

**Figure 2.** Compare the HFOWD (in blue) and VLSFO (in orange) from Asia to C.A, back and forth: (**a**) fuel consumption of the VLSFO is significantly higher than the HFOWD; (**b**) $CO_2$ emissions of the VLSFO are significantly higher than the HFOWD.

Figure 2b illustrates that the $CO_2$ emission from Asia to C.A. is 435.18 tons/day and 474.8 tons/day for HFOWD and VLSFO, respectively. As a result, the HFOWD can save 39.62 tons/day of $CO_2$ emissions. Similarly, the $CO_2$ emissions from C.A. to Asia are 421.98 tons/day and 467.02 tons/day for HFOWD and VLSFO, respectively. The HFOWD can save 45.04 tons/day of $CO_2$ emissions. On average, it turns out that HFOWD can save 42.33 tons/day (9.5%) of $CO_2$ emission in comparison with the VLSFO.

Under the same main engine output, 0.5% VLSFO consumes more fuel and produces more $CO_2$ emissions per metric ton than HFOWD. According to Alphaliner's statistics in July 2021, 85.2% of the world's approximately 5447 container ships used VLSFO, and 785 (14.4%) adopted HFOWD equipped with desulfurization towers, of which 7500~9999 TEU container ships account for the most significant proportion (there are 129 (16.4%)) [19]. According to the statistic usage of container ship fuel in 2021 and the above findings, the $CO_2$ emissions of container ships after IMO's 2020 sulfur cap are higher than before. Just as the results of Kontovas (2020) demonstrate [18], the total $CO_2$ emissions of marine fuel oil in 2020 will be 638.32 million tons, higher than the total $CO_2$ emissions of heavy fuel oil in 2019 of 618.6 million tons [18].

The results of this study are consistent with the conclusions of Ben-Hakoun et al. (2021) [10]. VLSFO creates more $CO_2$ emissions than HFOWD. There are two reasons why the main engine's daily fuel consumption of VLSFO is higher than the HFOWD. At first,

the supplier mixed the instability, poor lubricity, and poor compatibility of the impurified oil with the VLSFO. They are mainly composed of residuals and distillate components. As a result, the blended fuel reduces its inherent stability and compatibility. It causes waxy components, asphaltene precipitation, decreased viscosity, poor ignition, and fine catalyst content. Furthermore, high levels of catalyst fine content may cause abrasive wear to engine components [29]. Secondly, in investigating ways to meet IMO's requirements for sulfur restriction, much evidence has emerged indicating that the VLSFO also damages the engine. Adding lubricants and chemical additives can reduce the damage of VLSFO to the engine [30].

Therefore, the VLSFO may lead to the main engine wear [10], causing high fuel consumption and $CO_2$ emissions. In addition, the engine is still designed for high-sulfur equipment [10]; it's design does not account for low-sulfur oil. Therefore, we recommend that the fuel quality be prioritized to improve the quality stability of VLSFO [31]. In addition, if the same main engine continues to use low-sulfur fuel in the future, it is recommended to strengthen the leading engine equipment, such as using diamond-like carbon (DLC), multi-layer coating to reduce surface wear and improve self-lubrication [31], thereby improving VLSFO fuel efficiency.

### 3.3. Advantages and Weaknesses of Both Fuel Options

After the sulfur cap in 2020, sulfur oxides in the atmosphere decreased significantly [14], but $CO_2$ increased significantly [18]. Choosing the HFOWD involves facing the cost of adding desulfurization tower equipment to achieve the same level of desulfurization as VLSFO [17]. For fuel costs, taking the first half of 2021 as an example, VLSFO is higher than HFO, about USD 110/ton [19]. Another problem of VLSFO is poor fuel quality [29], resulting in equipment wear [10], which needs to be resolved. In general, the high calorific value of fuel saves fuel oil. However, VLSFO is a mixed fuel to meet the sulfur content of 0.5% m/m, in which lubricating oil and chemical additives are added, resulting in a high level of fine catalyst content. Consequently, the event causes the engine to wear. On the other hand, the engine utilized by VLSFO is specially designed for HFO, which is unsuitable for the VLSFO engine; thus, the equipment will wear out. Consequently, it increases fuel consumption and the overall $CO_2$ emissions that hasten global warming and climate change.

### 4. Conclusions

This study verifies a comparison of the fuel consumption and $CO_2$ emissions of HFOWD and VLSFO while ships are engaged in oceanographic navigation under the IMO sulfur restriction policy. The result is very successful. Both FOs reduce the sulfur oxides in the air, but the issue of $CO_2$ emission still exists. The FO consumption and $CO_2$ emissions of the VLSFO are higher than those of the HFOWD. That is due to poor fuel quality of VLSFO and the main engine equipment's wear. Consequently, with FO consumption at 130 tons/day for HFOWD and 141 tons/day for VLSFO, VLSFO shows an increases of 8.4%. $CO_2$ emissions are 429 tons/day for HFOWD and 471 tons/day for VLSFO, indicating 9.5% increase.

This task recommends strengthening the main engine material properties, such as piston rings and cylinder liners made of DLC, multi-layered coatings, and compound coatings that reduce wear and improve lubrication, and improve the quality of VLSFO. Furthermore, the current FOs can only initially achieve the goal of reducing sulfide emissions, and they cannot achieve the goal of carbon reduction. To achieve decarbonization in shipping, itis necessary to promote alternative low-carbon fuels and develop renewable energy fuels.

**Supplementary Materials:** The following supporting information can be downloaded at: https://www.mdpi.com/article/10.3390/app12199857/s1, Figure S1: Flowchart of the acquired data in the procedure; Figure S2: The gauge of measurement of fuel oil consumption on the board; Table S1: Tabulates the accuracy in the engine operational test by the original factory (provider); Table S2: The raw data the voyages.

**Author Contributions:** Conceptualization, S.-H.C.; data curation, H.-C.S.; formal analysis, H.-C.S.; F.-M.T. and C.-K.Y.; methodology, H.-C.S., F.-M.T., C.-K.Y., W.-Y.H., H.-P.P. and C.L.; resources, S.-H.C.; supervision, S.-H.C.; validation, C.L. and C.-K.Y.; writing—original draft, H.-C.S.; writing—review and editing, F.-M.T. and C.L. All authors have read and agreed to the published version of the manuscript.

**Funding:** This project is partially supported by the Higher Education Cultivation Program on the characteristics of marine research in 2022, National Kaohsiung University of Science and Technology, Taiwan, under grants No. 111E9010P01.

**Institutional Review Board Statement:** Not applicable.

**Informed Consent Statement:** Not applicable.

**Data Availability Statement:** The data presented in this study are available on request from the corresponding author.

**Conflicts of Interest:** The authors declare no conflict of interest.

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
