# Peer review of "Verification of Fuel Consumption and Carbon Dioxide Emissions under Sulfur Restriction Policy during Oceanographic Navigation"

_applsci, doi:10.3390/app12199857_

Round 1

Reviewer 1 Report

The reviewer appreciate the effort of authors to conduct a study on carbon emissions under sulfur restriction policy. Valuable practical cases were employed for comparative analysis. However, more voyage data and complete calculations are expected to make the Results more convincible.

The following comments are some instructions, which are supposed to be helpful for improving the manuscript.

ž   Section “1. Introduction” are recommended to be reorganized with a more clear and logical manner. Backgroundtopic to be discussedliterature review→existing problemwhat problem is this manuscript trying to solve, sth like that.

ž   Lines 55-66 on Page 2: The content expressed in this paragraph is repeated with the content of Table 1, which is meaningless. It is recommended to delete this paragraph.

ž   All acronyms and abbreviations including FO, HFOWD, VLSFO, IMO, etc., only need to be given the full name at the first occurrence, and then could be used directly in the following text. fix this problem through the full manuscript.

ž   Lines 129-136 on Page 4: The content expressed in this paragraph is repeated with the content of Table 2, which is meaningless. It is recommended to delete this paragraph.

ž   Lines 170-171 on Page 5: “At the same time, CCFuel it means the carbon content in fuel and heavy oil is 21.1 [27,28], and the unit is kg C/GJ.” Is the same value suitable for both HFO and VLSFO?

ž   Section “3. Results and Discussion3.1. Description analysis for both fuel options”, VLSFO has higher Heating value (kcal/kg), but more main engine fuel consumption (ton/day) (ton/day), which is confusing. A reasonable explanation is expected.

ž   The research Methods of this manuscript are not complicated. What the most important are the Materials, that is, the raw data about the voyages to be researched. The reviewer think it is necessary to demonstrate raw data in the form of graphs or tables to some extent. It is not very convincible for the current Table 3.

Reviewer 2 Report

The work presented for review is a kind of research report. Please highlight the scientific aspect of this work.

Some comments:

1.    Please highlight what this work brings new to this field of science?
2.    The authors investigate the influence of the type of fuel on its consumption and exhaust emissions. Please include the parameters of these fuels, LHV, density, etc.
3.    What measuring equipment was used, and with what accuracy?
4.    Is it possible to determine the accuracy of the engine's operational tests?
5.    Were both container ships powered by exactly the same engine and had the same exhaust gas cleaning systems? 

Round 2

Reviewer 1 Report

The reviewer believe the manuscript has been sufficiently improved to warrant publication in Applied Sciences.